# The Dynamic Behavior of Chromatin in Response to DNA Double-Strand Breaks

**DOI:** 10.3390/genes13020215

**Published:** 2022-01-25

**Authors:** Fabiola García Fernández, Emmanuelle Fabre

**Affiliations:** 1Institut Curie, CNRS UMR3664, Sorbonne Université, F-75005 Paris, France; 2Génomes Biologie Cellulaire et Thérapeutiques, CNRS UMR7212, INSERM U944, Université de Paris, F-75010 Paris, France

**Keywords:** chromatin, double strand break (DSB), chromatin dynamics, DNA damage response (DDR), genome integrity

## Abstract

The primary functions of the eukaryotic nucleus as a site for the storage, retrieval, and replication of information require a highly dynamic chromatin organization, which can be affected by the presence of DNA damage. In response to double-strand breaks (DSBs), the mobility of chromatin at the break site is severely affected and, to a lesser extent, that of other chromosomes. The how and why of such movement has been widely studied over the last two decades, leading to different mechanistic models and proposed potential roles underlying both local and global mobility. Here, we review the state of the knowledge on current issues affecting chromatin mobility upon DSBs, and highlight its role as a crucial step in the DNA damage response (DDR).

## 1. Introduction

To faithfully maintain genome stability and survive double-strand breaks (DSBs), cells display a DNA Damage Response (DDR) involving complex signaling networks that sense, signal, and repair DSBs. Inherited or acquired defects in the DDR were found in patients’ diseases, including immune deficiency, neurological degeneration, premature aging, and severe cancer susceptibility [1,2,3]. In addition to checkpoint activation, chromatin remodeling and repair itself, a process now considered to be part of DDR, increases the diffusion of damaged chromatin. Observed from yeast to mammals, this dynamic response constitutes a mechanism that seems to facilitate both the encounter of broken ends and the search for homology, therefore promoting genome stability [4,5,6,7]. However, uncontrolled DSB mobility can become deleterious under certain conditions, such as high amounts of DSBs or DSBs induced in TG-rich regions, triggering misrepair and subsequent chromosomal rearrangements [5,8]. In parallel to local movement, the dynamics of the chromatin away from DSBs are also altered, but the role of this global response is poorly understood [4,6,7,9,10,11,12,13,14,15]. Notably, over the last 20 years, several groups have hypothesized models explaining the potential mechanism and functions underlying both local and global motility due to damage. In this review, we present a clear description of the increase in chromatin dynamics in response to DSBs. To do so, we first present the most relevant features that influence the dynamic response of chromatin and then summarize the models proposed in the literature that shed light on the underlying mechanisms. Finally, we discuss the possible roles of this new DDR function in genomic integrity. 

## 2. The 3Rs of Double-Strand Breaks; Response, Repair, and Restoration: An Overview

The eukaryotic genome continuously experiences damage from endogenous and exogenous factors that can result in DNA base lesions, inter-strand crosslinks, or even single or double-strand breaks (DSBs). In humans, natural DNA insults can arise at least 10,000 times per cell per day. Much less frequent, but still important, are exogenous DNA insults triggered by radiation, chemotherapeutics, crosslinking agents, or radiomimetic compounds. Among all these different types of DNA damage, DSBs constitute the most detrimental lesions, having the potential to produce harmful genetic alterations including local modifications at the damaged DNA (insertions/deletions) or global chromosomal rearrangements such as translocations [16].

DSBs are rapidly detected by surveillance proteins that are recruited to the site of break, thus activating the DNA damage checkpoint. In addition to these DSBs sensing and signaling proteins, the checkpoint cascade also includes the recruitment of mediator and effector proteins that amplify the signal and arrest cell cycle to allow repair [17,18]. Among the principal checkpoint proteins, the mammalian polymerase PARP1 and the conserved KU protein are the most rapid sensing proteins, recruited within seconds after DSB induction (with a t 1/2 of only ~1.6 s after micro-irradiation). Other late-arriving sensor proteins include the MRN complex and the ATM and ATR kinases [19].

ATM (yeast Tel1) and ATR (Mec1) allow the recruitment and activation of several mediator and effector proteins, including 53BP1 (Rad9) and CHK1 (Rad53), respectively [20,21,22]. In addition to the DDR process, chromatin remodeling occurs as a result of checkpoint activation. Chromatin remodeling happens at the break site and in a genome-wide manner. It contributes to DSB signaling, but mainly facilitates access of the broken ends to the repair machinery. This remodeling step is mostly triggered by histone post-translational modifications (PTM) and the recruitment of specific remodeling complexes [23,24]. For example, one of the principal substrates of ATM and ATR is the variant histone H2AX (H2A in yeast), which becomes phosphorylated at serine 139 (or 129 in yeast) [25,26,27,28]. This PTM spreads several megabases around the break, but also *in trans* in mammalian topologically associated domains (TADs) or across yeast centromeres [29,30,31,32]. Phosphorylated H2AX (known as γH2AX) serves as a docking site for certain remodeling complexes, such as INO80 or SWR1, and checkpoint proteins such as 53BP1. These proteins mirror the phosphorylation mark and, in turn, can trigger downstream DDR factors [30,33,34,35], as reviewed in [36,37].

All the DDR processes not only lead to changes in chromatin structure, but also, as it is becoming increasingly clear, induce an increase in chromatin mobility. For example, the rapid recruitment of PARP1 leads to a transient chromatin relaxation that allows the recruitment of the upstream DDR factors ATM and MRX [38,39,40,41,42]. Likewise, the phosphorylation of H2A has been proposed in yeast to lead to a change in chromatin structure consistent with chromatin stiffening, which is necessary to boost DDR signaling [7,43]. 

Two major pathways are responsible for DSBs: non-homologous end-joining (NHEJ) and homologous recombination (HR). The HR pathway is considered the one predominantly used by single cell eukaryotes, such as yeasts *S. cerevisiae* and *S. pombe*, to repair DSBs. NHEJ, in contrast, is the more common DSB pathway in metazoans; however, the conservation of NHEJ factors throughout eukaryotes illuminates the use of this pathway even in yeast species (reviewed in [44,45,46]). Making the right choice of repair pathway ensures genome stability. This critical choice thus relies on several features, including the presence of a homologous sequence, cyclin-dependent kinase (CDK) balance, and DSB processing (reviewed in [47]. Under specific circumstances, when HR or NHEJ are deficient, alternative pathways such as single-strand annealing (SSA) and alternative end joining (Alt-EJ) may be induced [48,49]. However, these exert a cost on genome fidelity, as these pathways tend to be highly mutagenic.

Less is known about how the normal state of chromatin is restored after repair and how defects in this process affect genome integrity. However, the critical role of histone chaperones in restoring chromatin architecture after damage is beginning to be unveiled [50]. The following sections describe how chromatin mobility is also a key determinant during damage and during chromatin repair to restore genome integrity.

## 3. How, Where, and When to Measure Undamaged Chromatin Dynamics?

### 3.1. The How

Defining the dynamic behavior of chromatin fiber within the nucleus provides insight into how it explores nuclear space and how long it might take to make contact with functional molecular partners critical to DNA transactions, including replication, transcription, and repair [51,52]. 

Thus, to characterize DNA damage-induced chromatin dynamics, polymer physics is essential and it has been widely used to study the effects that damage can have on the spatiotemporal behavior of chromatin. The fact that yeast chromosomes in vivo behave like ideal polymers [43,53] has allowed many aspects of chromosome architecture and chromatin movement to be reproduced in silico by modeling yeast chromosomes as generic, semi-flexible polymers with a few sequence-specific rigidity constraints [53,54,55,56,57,58]. Thus, two main parameters govern the behavior of the chromatin polymer: compaction and stiffness. Chromatin compaction (C) can be defined as the number of base pairs per unit length along the fiber (bp/nm). Chromatin stiffness, or stiffening, is characterized by the persistence length (P), which can be understood as the distance over which the orientation between two monomers persists (in nm), such that stiffer fibers have a higher P. These two parameters are critical to the mechanical behavior of chromatin and dictate its movement during functional processes.

In vivo, 20 years of research have made it possible to follow a labeled genomic locus in real-time and to decipher the different parameters underlying the movement of chromosomes in living cells. To render a specific region of chromatin visible by microscopy as an individual spot, the best signal-to-noise ratio has been sought by different fluorescent labeling approaches coupled to microscopic imaging. Conventional lacO/LacI and tetO/TetR systems (often called FROS, for fluorescence repressor operator system) as well as innovative systems based on CRISPR (clustered regularly interspaced short palindromic repeat)-Cas9 (CRISPR-associated protein 9) are thus applied (reviewed in [59]). 

Based on the multiple time-lapse trajectories captured for a specific locus, it is possible to calculate its mean-squared displacement (MSD), allowing characterization of the amplitude of motion and the properties of the medium in which the tagged locus moves. An MSD curve represents the amount of space a locus has explored in the given time and its shape reveals the nature of chromatin motion. Accordingly, four principal types of motion have been described in the literature. First, the linear increase in the MSD with the time interval indicates the motion type of random walk, also known as Brownian motion; second, directional motion leads to an MSD curve deflected upwards while the third, confined motion results in a MSD curve deflected downwards; finally, when the force or structures restrict the motion, the motion is called anomalous sub-diffusion and does not show a simple confinement but is modulated in time and space with scaling properties. The MSD curve can be fitted with MSD ~ Dtα, where D is the diffusion coefficient (or sub-diffusion coefficient if sub-diffusive behavior) that represents the amplitude of DNA motion; t is the elapsed time; and α is the anomalous exponent. α can be estimated by fitting the slope of the MSD curve and indicates the diffusion behavior of the DNA locus and its possible interactions with its local environment. Thus, if α = 1, random diffusion is expected; α < 1 corresponds to a locus that rescans neighboring loci many times in a highly recurrent manner (sub-diffusive motion) and α > 1 indicates that the locus is able to often explore new environments (super-diffusive motion) [13,60,61]. Other motion parameters extracted from MSDs were used to characterize motion, such as the confinement radius (Rc) and the spring coefficient (Ks), which define the volume of the nucleus within which a locus moves and the external elastic forces applied to the DNA, respectively [61]. However, MSDs, as the name implies, are measurements of average movements and therefore do not provide information on different dynamic regimes. Thus, new methods for studying chromatin mobility can reveal transient directed movements of a locus that are masked by MSD analyses, such as the directional change distribution (DCD) method [62,63]. Interestingly, DCD analyses showed the nonlinear directional behavior of a locus after DNA damage induction [63]. Imaging techniques and microscopic tools at single-molecule resolution are also emerging, such as single-particle tracking (SPT) photoactivated localization microscopy, and provide insight into the location of individual molecules and their diffusive behavior [64,65].

### 3.2. The Where

All the tools described above have provided insight into how chromatin organization, from yeast to mammals, orchestrates its motion. Indeed, chromatin dynamics can be restricted by its interactions with nuclear substructures. For example, yeast chromosomes present in interphase the well-established Rabl configuration, where telomeres are clustered at the nuclear envelope (NE) by the end-binding complex Ku and the silent chromatin factor Sir4, and centromeres are attached to the spindle pole body (SPB) by microtubules (MT) [66,67,68,69,70,71,72]. Both tetherings restrict chromatin movement, as telomeres’ release or loss of centromeric anchoring increases chromatin dynamics of neighboring loci [11,58,61,73,74,75,76]. Interestingly, the bead-spring polymer model also predicted more decreased dynamics for peri-centromeric regions than for internal chromosome in regions in yeast, corroborating the effect of nuclear constraints on chromatin motion [56,58,75]. The motion of a chromosomal locus located far from constraint is thus differentially affected by its release, suggesting that chromosome conformation actually influences chromatin movement even in the absence of damage. Thus, the release of telomeric loci from the NE in mammalian or yeast cells showed profound effects on telomere mobility, as expected, but also on overall chromatin mobility [5,76,77]. In contrast to telomeres, recent studies in yeast have shown that although microtubules drive the dynamics of peri-centromeric regions, their depolymerization plays a relatively minor role on the mobility of loci distant from the centromere, a result predicted by polymer models [6,76,78]. 

As mentioned in the How section, one of the key parameters dictating chromatin motion is its compaction. Thus, measuring chromatin mobility in highly compacted regions, such as in mammalian heterochromatin, has revealed less dynamics than in regions with open chromatin [79,80,81,82,83]. A possible explanation is the tethering of heterochromatic loci to larger nuclear structures such as nucleoli, but also the state of chromatin, since forcing chromatin decompaction by inhibiting histone deacetylases or using deacetylase mutants enhanced chromatin dynamics [79]. Thus, the different dynamics observed according to the chromatin state and its nuclear localization already suggests that these two parameters may influence the dynamic response of damaged chromatin.

### 3.3. The When

In addition to knowing the environment of the tracked locus, basal DNA mobility can be altered by changes in chromatin organization resulting from various processes, from cell cycle progression to certain external and internal stimuli such as growth conditions. Indeed, early studies suggest that the movement of a given locus changes significantly with the phase of the cell cycle [73]. Tracking the dynamics of different chromatin loci throughout the cell cycle in yeast showed that all loci were highly mobile in G1 [6,73,84]. Likewise, mammalian cells in G1 exhibit the highest level of dynamics [79,85]. In contrast to what occurs in G1, chromatin dynamics in the S phase are limited due to DNA replication and, more specifically, to the involvement of the cohesin complex, which mediates the connection of sister chromatids after replication [6,73,84]. Accordingly, both the disruption of DNA replication and the mutation of one of the cohesin complex subunits led to an increase in chromatin mobility [6,73,75]. Finally, growth conditions must also be considered when measuring chromatin dynamics. Several studies have shown that chromatin mobility can be affected by the carbon source and temperature or pH of the culture medium used [86].

## 4. Chromatin Dynamics as a Part of the DNA Damage Response

In recent years, several studies have attempted to characterize the nature of increased chromatin mobility observed upon damage, using different DSB induction methods coupled to the microscopic approaches mentioned above. For example, targeted induction systems, including the HO endonuclease in yeast, TALE nucleases, the CRISPR/Cas9 system, homing endonucleases (I-SceI), and restriction enzymes, allow for the induction of a single DSB (or multiple DSBs) at given positions in the genome. These methods are useful for understanding the local movement of chromatin surrounding the damage and the overall movement of undamaged chromatin, if any (reviewed in [87], Figure 1).

Alternative methods such as ionizing radiation, cross-linking agents, radiomimetic compounds, and localized laser micro-irradiation induce 1–10 DSBs throughout the genome (~4 DSBs per yeast nucleus after 40 Gy of γ-radiation [88]; ~1 DSBs per mammalian nucleus after 2.75 Gy of γ-radiation [5]). Although the position of these DSBs is not precisely known, this is a way to study the mobility of undamaged (global) chromatin, given the low probability of tracking a damaged locus by this means. Intriguingly, the greater the number of induced breaks in a genome, the greater the mobility of its chromatin [4,5]. However, the ability of cells to survive many damages is challenging. This latter point suggests the importance of maintaining limited mobility during repair. 

### 4.1. Local Chromatin Mobility after DSB

The induction of both single and random DSBs was associated with enhanced local mobility, as evidenced by tracking a labeled locus near the DSB site or labeled repair focus after DSB induction. For example, in yeast, a tagged locus at 4–10 kb from the HO cutting site at MAT locus revealed increased chromatin mobility after HO induction [6,8,11,15,61,89]. Similarly, the induction of I-*Sce*I endonuclease at different genomic positions (*URA3*, *ZWR1* genes) enhanced the mobility of adjacent labeled loci [4,13,90]. On the other hand, foci that formed after zeocin treatment, labeled with the repair protein Rad52, show increased mobility [90]. The increase in local chromatin dynamics in all of these studies was measured after a long period of DSB induction, probably focusing on the later stages of the DDR. Accordingly, the increase in local chromatin dynamics observed in yeast is coupled with an expansion of the nuclear area explored by the damaged locus (40% more than in the undamaged state), possibly enabling homology searching [4,13,90]. Consistently, measuring dynamics during the initial phase of resection (5–40 min) shows a transient reduction in local chromatin mobility [91], highlighting different chromatin states along the DDR. Furthermore, during the S phase, neither spontaneous DSBs that are repaired by sister chromatid exchange nor endogenous DSBs induced by HO during a physiological mating-type switch, appear to trigger changes in local chromatin mobility [84]. Enhanced chromatin dynamics seem therefore unnecessary when a homologous template is available nearby.

As in yeast, tracking the mobility of irradiation-induced damage foci (IRIFs) in different mammalian cell types revealed increased dynamics compared to the mobility of chromatin before damage [79,92,93,94,95,96]. Similarly, increased local dynamics were demonstrated at uncapped telomeres (DSB-like structures), as well as at telomeres underlying the alternative lengthening pathway (ALT) [5,97,98]. Yet, other studies indicated that IRIFs or DSBs induced by endonucleases do not trigger increased local chromatin dynamics [40,99,100,101]. However, some parameters, such as the phase of the DDR or the time scale, were not controlled in these later studies.

### 4.2. Global Chromatin Mobility after DSB

The term global dynamics was first evoked by measuring the movement of an undamaged chromosome after the induction of a single DSB in its homologous counterpart in yeast diploid cells [4]. Similarly, an HO-induced DSB on yeast chromosome IV of haploid cells, increased the movement of a tagged locus on chromosome V (*MAK10*), although to a lesser extent than the local mobility [11]. Likewise, an HO-induced DSB at the *MAT* locus on yeast chromosome III increases the mobility of an ectopic region in chromosome VI (*MET10*) [6]. A similar effect on global mobility was also observed when inducing random damages using zeocin [7,9,10,12] or IR [4,5,13,14]. 

In contrast to yeast, global chromatin mobility was not widely explored in mammals. However, it was shown that random IR-induced DSBs increased the dynamics of undamaged tagged telomeres, but to a lesser extent than damaged ones [5]. Likewise, by following the dynamics in vivo of GFP-tagged mammalian histones H2B, another study discovered that chromatin moves in a genome-wide manner after DSB induction by cytotoxic drugs [102]. Despite this latter evidence, other studies failed to find any change in undamaged chromatin [79,97]. A more systematic study, taking into account the type of damaged chromatin and its organization in the nucleus is therefore necessary to have a clearer picture of the existence of global mobility in mammals.

### 4.3. Type of Chromatin Motion after Damage

Motion derived from most MSD-derived analyses in yeast and mammals after DSB induction has indicated a sub-diffusion behavior [7,11,14,43,56,61,62,79,89,97]. These analyses also described an increase of the sub-diffusive exponent α after DSB induction, suggesting modifications in the chromatin structure. Accordingly, different polymer models, as well as high-resolution microscopy, have confirmed these chromatin changes and in addition proposed that chromatin structure becomes more flexible [56], expanded [61], or rigid [13,43].

Recent additional studies have revealed complex mobility profiles of damaged DNA, encompassing periods of sub-diffusive and directed motions [63,98,103,104]. For example, by using modified MSD analyses, damaged telomeres exhibited sub-diffusive motion before moving directionally upon capture of the homologous template [98]. Similarly, DCD analyses of a damaged locus in yeast revealed transient non-linear directional motion of damaged DNA [63]. Studies of dynamics obtained in yeast after damage are compiled in Table 1.

## 5. Models Explaining DSB-Induced Chromatin Mobility

The observed increase in local and global dynamics after damage seems to be driven by different mechanisms, such as intrinsic modifications of chromatin due to its altered structure, by the loading of the DDR machinery or as extrinsic forces mediated by the nuclear architecture (Figure 2).

### 5.1. Chromatin Fiber Modifications

As seen above, the different chromatin landscapes during DDR are characterized by structural changes such as stiffness, compaction, and condensation. Early studies have shown that random DSBs induced either by UV radiation or by bleomycin lead to an increased sensitivity of chromatin towards the MNase digestion, indicating a higher accessibility at the nucleosomal level [27,105], correlated with chromatin de-condensation [39,40,106,107]. Besides an increase in mobility, intra-chromosomal distances measured between different pairs of labeled loci (separated by 200 Kb) along a yeast chromosome arm, increased after DNA damage, indicating changes in the fiber itself [43]. This change in structure was consistent with a more rigid chromatin according to yeast chromosome polymer simulations, in which the biophysical parameters C and P were modulated [43,53]. This most likely interpretation, though surprising, was also coherent with the increase after DNA damage in the anomalous subdiffusive exponent, another parameter indicating structural changes in chromatin [43,61]. However, by coupling polymer models to experimental data, it was predicted that the increased α values observed after damage can also be interpreted as polymer decompaction [61]. Curiously, superresolution images of a labeled locus after random damage corroborate both global stiffening and decompaction of chromatin after damage [43,61]. 

To reconcile these observations, it is important to consider the concept of condensation and compaction, which is the number of nucleosomes per unit volume (nucleosome volume density) or per unit length (linear nucleosome density), respectively. By carrying out genome-wide nucleosome mapping, Gasser’s lab showed a global decrease in nucleosomes after DNA damage, corroborating chromatin decompaction [12]. However, decompaction does not characterize the structure of the chromatin fiber itself (but rather the total volume of nucleosomes), whereas rigidity does, indicating that these two chromatin modifications can indeed coexist. It will definitely be of great interest to compare the data from polymer models with those from high-resolution images of chromatin fibers upon damage.

All of these structural changes in chromatin were related to increased chromatin dynamics. Notably, artificial reduction by the transcriptional inhibition of yeast histones H3 and H4 in the absence of damage triggers an increase in global chromatin motion, comparable to the dynamics induced by zeocin [12]. Similarly, the induction of an HO-targeted DSB in the yeast MAT locus results in both local histone eviction and a local increase in chromatin dynamics [61]. On the other hand, chromatin stiffening was found to be consistent with the repulsion between nucleosomes due to the negative charges promoted by γH2A(X) as shown in vitro [108]. These observations pave the way for the involvement of chromatin mobility in DDR, as developed in the following paragraphs.

### 5.2. DDR Implication in Chromatin Mobility

#### 5.2.1. Remodeling Machinery and Post-Translational Histone Modifications

Global histone loss or histone H2AX phosphorylation upon damage are likely to enable DNA accessibility to repair. Histone removal, in particular, is triggered by the remodeling machinery loading. Recently, a new mass spectrometry-based approach developed to analyze the chromatin-associated proteome after zeocin damage, confirmed the partial loss of histone H1 and core chromatin along with chromatin remodeling machineries [15]. This method also allowed observation of the recruitment of some ubiquitin-conjugating factors (Rad6, Bre1, Pep5, Ufd4, and Rsp5), which contribute both to histone depletion [15].

It is interesting to note that the remodeling complexes INO80 and SWR1, which dislodge or exchange histones, respectively, are involved in chromatin motion after damage in yeast. Whereas INO80 appears to be partially required to increase local mobility, its disruption completely abolished global mobility [89]. In contrast, the SWR1 complex appears to mainly modulate local dynamics, since a tagged locus at 2.7 Kb from a DSB site was found to be less mobile in Δswr1 mutants [89]. These opposing effects of INO80 and SWR1 complexes on dynamics could explain their opposite roles in DDR, since SWR1 promotes NHEJ while INO80 facilitates HR [109]. Consistently, the histone H2AZ variant, which is incorporated by SWR1, also prevented local chromatin dynamics of the same tagged locus [89]. Interestingly, as chromatin remodeling requires metabolic energy, it was expected and confirmed that chromatin remodelers mediate mobility in an ATPase activity-dependent manner [110]. Accordingly, ATP depletion by ATP synthesis inhibition and sodium azide significantly (34%) decreased the mammalian IRIF displacement [79]. 

As seen above, the long-scale spreading of γH2A(X) enhances chromatin mobility, as shown in yeast [7,43]. Similarly, in mammals, wide regions of γH2A(X) chromatin displayed changes in chromatin fiber interpreted as a re-condensation [38,39,40,41]. Remarkably, these chromatin modifications mediated by γH2A(X) on such a large scale seem to contribute to the motion of the chromatin fiber also in mammals [40,82]. Thus, when γH2A(X) is suppressed by targeting its specific substrate (S129 in yeast or S139 in mammals) or the kinases (Mec1/Tel1 or ATR/ATM), global chromatin dynamics after damage decrease significantly [9,14,90,92,94,96,97]. Consistently, a S129E mutant that mimics phosphorylation of S129 in yeast, shows an increase in chromatin mobility in the absence of damage [7]. Besides phosphorylation, other PTMs have been involved in chromatin dynamics. For example, inhibiting histone methylation, preventing histone acetylation, or blocking histone acetyltransferases by using 5-azacytidine, trichostatin A, and curcumin, respectively, induced a significant reduction (20%) in mammalian IRIF mobility [79]. Collectively, this evidence in yeast and mammals argues that in response to DNA damage, chromatin remodeling is involved in chromatin mobility.

#### 5.2.2. Checkpoint Machinery

Checkpoint activation indirectly modifies chromatin, since chromatin remodeling occurs only if the damage is accurately signaled by the checkpoint cascade. Studies in yeast and mammals have described the involvement of the checkpoint proteins 53BP1/Rad9 and CHK1/Rad53, in addition to ATR/Mec1 and ATM/Tel1, in both local and global chromatin dynamics, but in different conditions.

Locally, the enhanced mobility of a locus 2.7 kb from an induced I-SceI cutting site in yeast was dependent on the kinases Mec1 and Tel1 and the mediated protein Rad9 [90]. Studies in mammalian cells similarly show increased local dynamics of IRIFs and unprotected telomeres mediated by ATM [5,94,96,97]. Regulation by 53BP1 is more versatile as the increase in unprotected telomere dynamics depends on it, but not IRIF dynamics [5,79,97]. ATM/Tel1 involvement in local mobility thus appears highly conserved from yeast to mammals, with the involvement of 53P1/Rad9 being apparently dependent on the type of damage.

Globally, artificial activation by a GAL promoter of yeast Ddc1 and Ddc2 kinases, which form dimers with Tel1 and Mec1 respectively, enhanced global chromatin dynamics similarly to zeocin treatment [9]. According to this, in yeast cells, treatment with caffeine, which inhibits Mec1 and Tel1 kinases, suppresses the increase in global mobility observed upon damage, as well as the deletion of Rad9 [7,9,14]. Moreover, the Cep3 and γH2A(X) targets of Mec1 were also implicated in enhanced global dynamics after a single or random DSB induction, respectively [7,11,43,92]. Curiously, the effector protein Rad53 seems to mediate global dynamics without affecting local dynamics.

Thus, an intimate interaction between the checkpoint and remodeling mechanisms appears to govern the signaling events that produce global chromatin mobility. However, how they cooperate to control motility has not been elucidated, perhaps a chicken-and-egg situation.

#### 5.2.3. Repair Machinery

Besides the checkpoint machinery, several repair factors were found indispensable for chromatin movement after DNA damage. As seen above, there is a transient decreased local movement after damage due to DSB resection [4,91]. Downstream resection, recombination proteins such as Rad52, Rad54, and Rad51 are required for increased DSB motion in mammals and yeast. For example, in mammals, Rad51 knockdown limits the increase in mobility of damaged telomeres [98]. Similarly, the roles of Rad51 and Rad54 in local mobility are documented in many yeast studies [4,13,14,63,90]. The role of Rad51 in global mobility seems, however, to be dependent on cell ploidy, because Rad51 promotes the IR-enhanced mobility of undamaged loci in diploid cells [4,14], but becomes dispensable after zeocin treatment in haploid cells [9,90]. The discrepancy between these studies could be explained by the hyperactivation of Rad51 in diploid cells [9]. Interestingly, the yeast Δrad51-II3A mutant, which is defective in strand exchange but shows normal ssDNA binding, leads to enhanced chromatin dynamics in diploid yeast after IR, suggesting that Rad51 controls global chromatin mobility by its association with the ssDNA [14]. Furthermore, Rad52 is involved in enhanced global chromatin dynamics exclusively after its binding to Rad51, indicating that in the absence of a proper presynaptic Rad51 filament, global mobility is limited by Rad52. These results provide evidence for a regulatory circuit between Rad51 and Rad52 recombinases in the control of global mobility after random damage [14].

### 5.3. Extrinsic Forces Mediated by the Nuclear Architecture

Centromere anchoring and telomere tethering, typical of the Rabl yeast chromosome configuration, are challenged upon DNA damage. In response to a targeted DSB, it was thus first reported that increased chromatin motion arises from the loss of chromosomal anchors, specifically centromere and telomere tethering [11]. Accordingly, the clustering of GFP-tagged kinetochore protein Mtw1 or cohesin protein Smc3 was affected after DSB induction [11,76]. In accordance, zeocin exposure also led to an increase in 3D distances between the SPB and the centromere of chromosome IV [7,43]. Nevertheless, centromere detachment was not observed under zeocin treatment [6,76] and detachment of a centromere from its nuclear microtubule, by forcing transcription through the centromere, curiously does not induce chromatin mobility after DSBs [6,76]. Thus, although changes in the peri-centromeric region after damage were observed, the question of the role of centromeric tethering to the SPB in chromatin dynamics remains.

In addition to tethering, a crucial role for cytoskeletal components in local and global enhanced chromatin mobility after damage was highlighted [5,61,62,63,76,82,111,112]. Unexpectedly, it appears that cytoskeleton components such as actin and its regulators are found in the nucleus (reviewed in [113,114]). On the other hand, yeast nuclear microtubules, emanating from the SPB, have long been known. Recent evidence indicates that all these cytoskeletal components are involved in chromatin mobility when damaged [82,112]. In addition, crosstalk between the classical cytoskeleton and chromatin has also been implicated in chromatin dynamics upon damage. For instance, yeast cells subjected to random DSBs exhibited elevated levels of nuclear microtubule filaments, as seen by tracking simultaneously CFP-Rad52 foci and GFP-Tub1 (α-tubulin protein, [63]). These structures named DIMs, for DNA damage-inducible intranuclear microtubule filaments, resulted from microtubules and kinetochore components. Interestingly, super resolution images showed that DIMs are able to capture/mobilize damaged DNA in a non-linear directionality in order to repair [63]. In agreement with this, actin or microtubules perturbations in yeast treated with latrunculin or nocodazole, respectively, impede both local and global increase of chromatin dynamics upon damage [61,76,111]. Likewise, disrupting the yeast kinesin-associated proteins, Kar3 and Cik4, abolished the enhanced local dynamics observed in WT cells after an HO-induced DSB [62]. In mammals, the enhancement in chromatin mobility after random DSB was linked to the LINC complex (Linker of Nucleoskeleton and Cytoskeleton), a protein complex that connects cytoskeletal microfilaments and microtubules with the nucleus interior. This complex was indeed implicated in the enhanced chromatin dynamics seen at unprotected telomeres [5]. These different observations suggest that oscillatory forces mediated by cytoskeletal components on the nucleus can regulate chromatin movement within the nucleus from yeast to mammals.

### 5.4. Cohesin-Mediated Loop Extrusion

While it is not clear yet if chromatin loops are related to global mobility, sufficient data now suggest that cohesin-mediated loop extrusion plays an important part in chromatin movements around DSBs. Recent evidence in yeast and mammals have shown that, following damage, chromatin folds into cohesin-dependent loops that influence dynamic inter- and intra-chromosomal contacts [115,116,117,118,119]. For example, in yeast, following an HO-induced DSB, chromatin flanking the DSB is proposed to prevent cohesin progression along the chromosome. Subsequent folding of chromatin into loops of ~20 kb long promotes increased *cis*- contacts while inhibiting *trans*- contacts [118]. 

Similarly, mammalian TADs are formed by cohesin-mediated chromatin loops. Cohesins are found enriched at TAD boundaries due to their interaction with the border protein CTCF [119,120,121,122]. CTCF and cohesins form a dynamic fast-exchange complex that makes TADs highly dynamic [121]. The induction of a DSB within a TAD results in loop extrusion triggered by ATM-dependent cohesin accumulation and promotes H2AX phosphorylation during loop extrusion in a directional and progressive manner [119]. 

## 6. Potential Functions of DSB-Induced Chromatin Mobility

Before addressing the roles of increased chromatin mobility during the DDR, it is worth noting that the entire genome does not behave similarly in DSB repair and that repair rates may be affected by the genomic position of the damage. Different genomic domains are thus classified as repair-repressive or repair-prone domains. While the nuclear pore complexes and the nuclear periphery are documented as repair-prone domains [80,103,123,124], the repair-repressive domains correspond to heterochromatin, ribosomal DNA, and transcribed regions [82,125]. Thus, DSB mobility was found to be required for DSBs induced in repair-repressive domains to move out of these regions to reach repair-prone domains [61,103,123,125]. Evidence for chromatin mobility has also been provided by the observation of “repair centers” in which Rad52, a repair protein bound to the DSB, can cluster and probably repair. Early studies in yeast have shown that two DSBs induced in different chromosomes are colocalized with a single Rad52 focus [126]. It was suggested that the Rad52-focus might facilitate recombinational repair [4,126,127]. A more recent study confirmed the formation of Rad52 foci using super-resolution microscopy and further characterized Rad52-diffusion behavior within the foci [127]. Similarly, it was observed that γH2A(X)domains and 53BP1 foci formed in irradiated mammalian cells clustered, allowing the recruitment of the HR machinery [79,92], as confirmed by Hi-C when inducing DSBs at defined loci in the human genome [82]. Interestingly, DSB clustering requires chromatin mobility driven by actin-related proteins and the formin actin regulator [82]. This genome-wide mapping of long-range contacts revealed that DSBs cluster only when they are induced in transcriptionally active genes (repair repressive domains). This observation suggests that clustering depends on the nature of the damaged loci and deserves further investigation. However, the function of DSB clusters as HR centers was recently challenged in yeast where three DSBs formed two to three foci, rather than one, that could dynamically fuse and coalesce. The fact that these foci could form in a Δrad52 strain, defective in HR, further suggests that these foci may not act as “reparosome” centers [128]. Nevertheless, these DSB relocation and clustering events require precise regulation of chromatin mobility in time and space. 

Functionally, nuclear chromatin organization and mobility appear to act together to promote genome stability (Figure 3), by helping repair accessibility or modulating the repair pathway choice. In the past decade, increased chromatin mobility at the DSB (local) has been involved in certain DDR events, such as DSBs clustering, re-localization to repair-prone domains, or homology searching. The role of chromatin mobility elsewhere (globally), on the other hand, remains poorly understood, although some studies link it to the different repair pathways. Importantly, since the amount of damage modulates chromatin mobility, mobility may also restrict DDR functions, as uncontrolled dynamics might be detrimental to cell fate.

### 6.1. To Relocalize DSB

DSBs induced in repair-repressive domains must reach the domains that will ensure repair, which requires at least local DSB mobility. Thus, studies in yeast and mammals have shown that enhanced local dynamics of a DSB targeted in rDNA enabled its exclusion from the nucleolus [125,129,130,131]. Similarly, the enhanced local dynamics of DSBs observed after IR exposure or I-*Sce*I induction in fly heterochromatin allowed the relocalization of damaged foci to the nuclear periphery [80,103,123,124]. Moreover, both nucleolar exclusion and relocation of DSBs from heterochromatin were dependent on cohesin, nuclear actin, and chromatin compaction. Importantly, altered chromatin mobility following DSB damage or relocation increased the rate of chromosomal rearrangements, highlighting the role of mobility in proper repair [80,103,125].

DSBs can also be relocalized when DSBs are said irreparable because there is no homology or the DSBs are persistent (due to permanent damage induction). For instance, it was shown in yeast that in the absence of homology, Rad51 remains bound at the broken DNA end, indicating persistent homology searching, ultimately leading to relocation of the DSB to the nuclear periphery, prone for repair [132,133,134]. Likewise, persistent DSBs induced in both budding and fission yeast were observed to move towards the nuclear periphery, at the inner nuclear membrane or nuclear pores [63,89,132,135]. This is also the case for highly mobile Rad52 foci induced by MMS treatment, which reached the nuclear periphery through damage-inducible intranuclear microtubule (DIM) filaments [63]. Similarly, a persistent DSB induced in different chromatin regions by restriction enzymes is relocalized to the nuclear periphery, depending on SWR1, kinesin, and SUMO (for small ubiquitin-like modifiers) metabolism [8,63,132,134].

### 6.2. To Amplify the DNA Damage Response

Increased chromatin dynamics after DSB induction have been shown to increase the local processivity of certain DDR factors, including ATM, 53BP1, and RIF1, thereby ensuring efficient DDR where chromatin loop formation may have a role to play [115,116,117,119,122]. For example, several groups have observed that local movements within a TAD through chromatin loops bring distant nucleosomes into spatial proximity to ATM kinase, ensuring the phosphorylation of γH2AX across the TAD [115,117,119,122]. Therefore, the deletion of the CTCF chromatin binding protein reduces both chromatin movement and γH2AX spreading in a TAD [121,122]. Furthermore, loop extrusion is not observed in ATM-deficient cells, indicating the presence of a mechanism to protect the integrity of genome structure during DNA damage repair [117].

Similarly, the ability of yeast Mec1/Tel1 kinases to phosphorylate not only at the DSB, but also in undamaged regions *in trans*, further supports the importance of chromatin mobility in generating DDR platforms, perhaps through chromatin loops [31,32]. Accordingly, *in trans* propagation of γH2A in peri-centromeric regions is consistent with the presence of chromatin loops in centromeric regions, reviewed in [136], although Mec1 propagation in 3D, as reported through mathematical modeling, does not necessarily require loop formation [137].

### 6.3. To Promote Repair by HR or NHEJ

Chromatin mobility in response to DSB was linked to the two main repair pathways: NHEJ and HR. First, chromosome mobility was suggested to be a means for promoting reconnection and joining of double-stranded extremities, i.e., NHEJ [5,7,97,138]. The increased roaming of deficient mammalian telomeres or the DSB-enhanced mobility in irradiated or bleomycine-treated mammalian cells facilitate NHEJ in a 53BP1 and DNA-PK dependent manner [5,97,138]. Likewise, the mobility mediated by Rad9 in yeast exposed to zeocin seems to facilitate NHEJ [7]. Interestingly in this later study, it was shown that the increase in chromatin dynamics due to H2A phosphorylation mimicry was favorable to local NHEJ repair [7].

On the other hand, since the spatial proximity of DNA compounds seems to influence repair by homologous recombination, chromatin mobility is likely to help repair a broken chromosome if successful recombination is not immediately possible. According to this hypothesis, for events where recombination is feasible, such as at the replication fork or for a mating-type switch, chromosome mobility is not necessary and therefore not affected [84,90]. In contrast, increased chromatin mobility allows DSB repair when its homologous donor sequence is located on another spatially distant chromosome; the further the donor is from the break, the more mobility is required [11]. Moreover, the concomitant enrichment of Rad51 and γH2A in a global manner is a direct consequence of homology search that can potentially be explained by increased chromatin mobility [32]. Accordingly, both factors are necessary for both global and local mobility. Thus, a model was formulated from these observations in yeast cells, proposing that mobility of the damaged ends, combined with a global mobility, act like a “needle in a ball of yarn”, enhancing the ability of the break to traverse the chromatin meshwork and thus facilitating homology searching [13]. However, the role of local mobility in homology search has recently been challenged by a study in yeast in which altering local, but not global, mobility did not affect either survival or homology-mediated strand invasion [6].

### 6.4. To Avoid or Induce Translocations

The notion that mobility promotes repair by NHEJ is challenged by the fact that DSB ends must remain together and, therefore, enhanced mobility could disrupt their tethering. However, changes in DNA mobility act as a double-edged sword; they can promote precise repair, but in some cases, they lead to potentially mutagenic DNA repair events and are the source of chromosomal translocations. For example, increased mobility promoted by an induced DSB at the yeast MAT locus leads to NHEJ repair. However, the insertion of TG-rich repeats near the cleavage site in MAT allows uncoordinated movement of the break ends, which leads to homology invasion on another chromosome and results in a translocation event [8]. Accordingly, a study in mammals showed that extensive local movement promoted by simultaneous-induced DSBs correlated with translocations and deletion events [139,140]. Conversely, increased dynamics mediated by H2A phosphorylation or its involved kinases not only promotes end joining but also prevents ectopic repair [7,140,141,142], indicating that this PTM is one of the factors limiting chromosomal translocations. It is therefore crucial to know to what extent chromatin dynamics can be beneficial and to decipher their threshold point, i.e., when it becomes detrimental.

## 7. Conclusions

The evolutionary conservation of chromosome mobility after DNA damage highlights its role as a key regulator for chromosome integrity. Recent advances in microscopy and DSB-induced systems allowed a more precise characterization of chromatin mobility in the context of damage. As a result, we have a better understanding of the factors that strongly influence chromatin movement, such as the phase of the cell cycle, the physical characteristics of the chromatin region observed, and the specific tools used to measure and analyze it. However, the mechanisms and functions of increased mobility remain rather intriguing. For both local and global mobility, the precise contributions of intrinsic chromatin modifications and/or external molecular motors require further validation. Similarly, while the role of local dynamics seems to be increasingly clear in the different repair pathways, more approaches are needed for testing the relevance of global mobility for efficient DNA repair. Some key questions remain: How the different changes in chromatin organization due to damage alter its dynamics? What is the nature of the chromatin modification underlying chromatin mobility and is it related to a specific DDR? What are the conditions under which the dynamics become detrimental? Chromatin mobility, a new parameter of the DDR, has not yet revealed all its mysteries.

## Figures and Tables

**Figure 1 genes-13-00215-f001:**
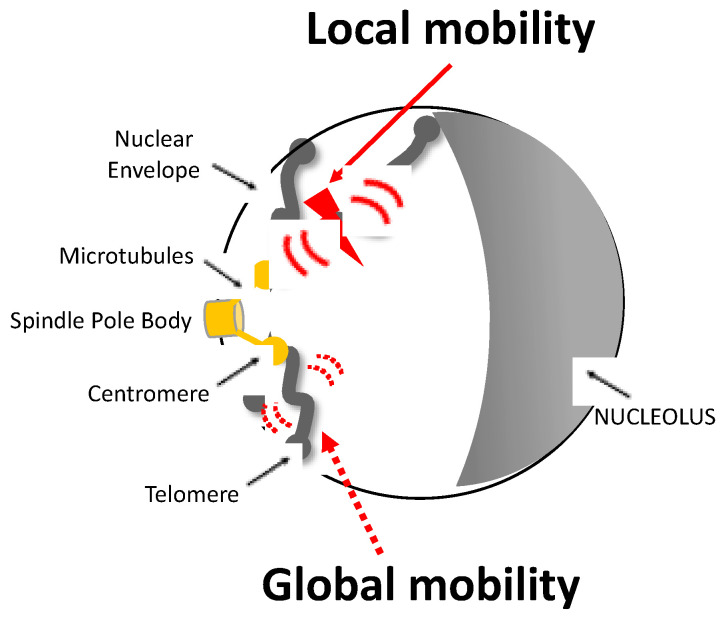
Representation of local and global mobility after a single DSB in a yeast nucleus. The Rabl configuration of a haploid yeast cell is illustrated: chromosomes are attached to the spindle body (SPB) by their centromere (CEN) via a nuclear microtubule (MT) and to the nuclear envelope (NEV) by their telomere (TEL). The induction of a DSB, represented by the red flash, triggers an increase in the mobility of nearby chromatin known as local mobility and that of other chromosomes known as global mobility. Local and global mobility are symbolized by large red and dashed red arrows, respectively.

**Figure 2 genes-13-00215-f002:**
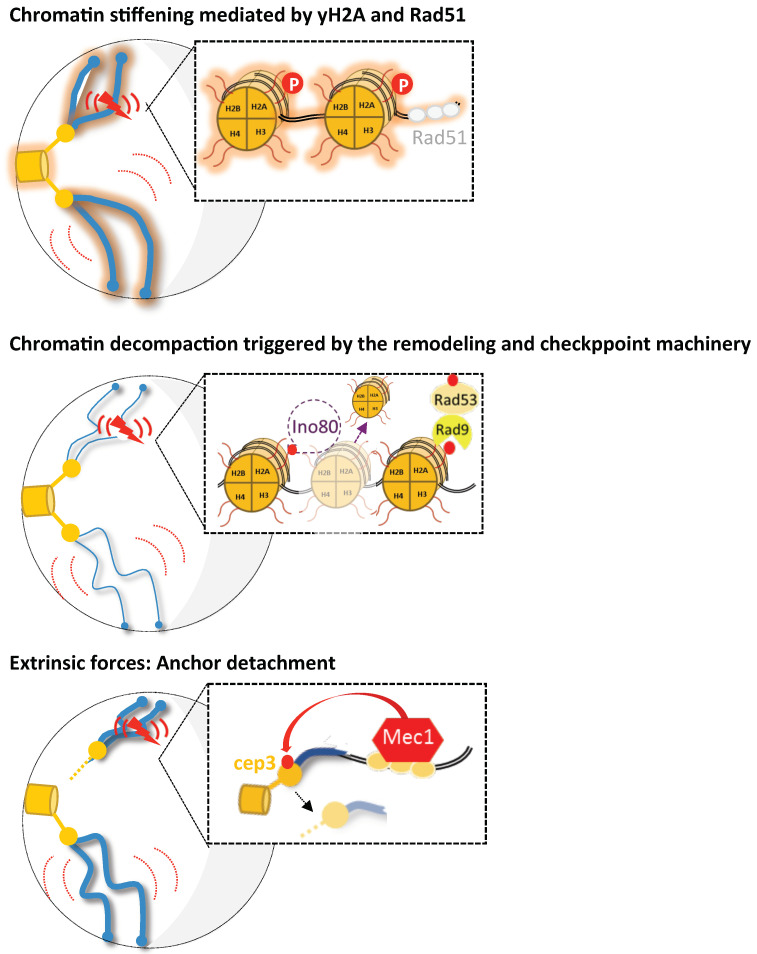
Representation for the various mechanisms proposed to increase chromosomal mobility upon DNA damage in budding yeast. Under damaged conditions (DSB is represented by the red flash), three possible scenarios of enhanced chromatin mobility are shown (global mobility is symbolized by large red curves and local by short red curves). First, the proposed mechanisms for global and local mobility include chromatin stiffening mediated either by H2A phosphorylation or Rad51 nucleofilament formation (as proposed in [10,13], respectively). A second model proposes chromatin remodeling as the mechanism underlying enhanced local and global chromatin mobility. Checkpoint signaling triggers INO80-C-dependent histone loss, thus leading to subsequent chromatin decompaction (represented by thinner chromosomes) [9,12,90]. Finally, in Strecker et al., 2016, it was proposed that Mec1 activation leads to the phosphorylation of the kinetochore protein Cep3. Thus, Cep3 phosphorylation would modulate the kinetochore/centromere attachments and lead to increased global and local dynamics. Drawing inspired from [14].

**Figure 3 genes-13-00215-f003:**
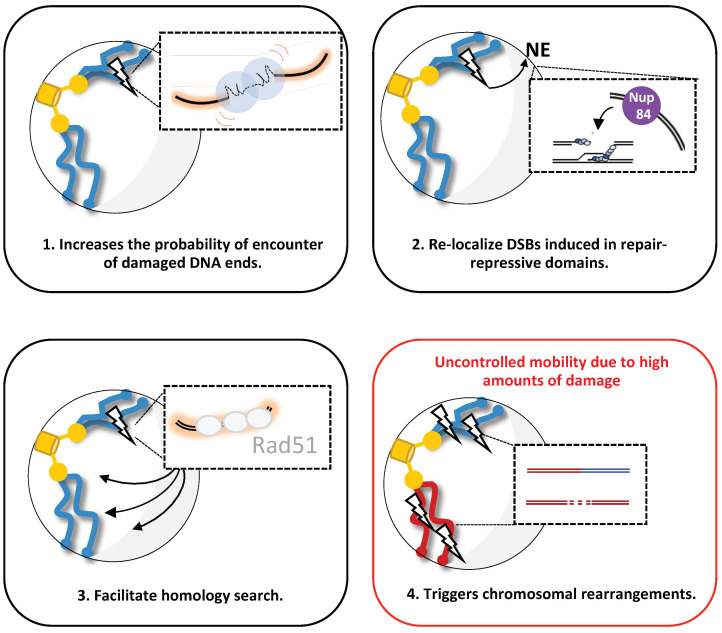
Proposed function of increased mobility upon DSB(s) induction in budding yeast. Four possible scenarios are shown: (**1**) after a DSB induction, enhanced chromatin mobility (red curves) facilitates the probability of accurate rejoining of DSBs, promoting c-NHEJ. (**2**) After DSB induction in repair-repressive domains or persistent DSB, enhanced chromatin mobility mediates DSB relocalization to the nuclear envelope (NE), specifically Nup84 at the nuclear pore complex, where it can be repaired. (**3**) After DSB induction, enhanced chromatin mobility facilitates homology searching done by the stiffer nucleofilament Rad51, allowing HR. (**4**) Representation of a specific condition in which high amounts of DNA damage trigger uncontrolled chromatin mobility, favoring chromosomal rearrangements.

**Table 1 genes-13-00215-t001:** Summary table of yeast studies exploring chromatin dynamics upon DSB induction in budding yeast.

Study	Organism	Type of Damage	Local Mobility	Global Mobility	Type of Motion	Involved Proteins	Function of Dynamics	Methods
[61]	Yeast (Hap.)	HO @ MAT locus	increased	-	Sub-diffusive	Ino80	DSB relocalization	FROS, time-lapse fluorescence, super-resolution imaging, and polymer model (β polymer model).
[15]	Yeast (Hap.)	Zeocin (250–300 µg/mL); HO @ MAT locus	increased	increased	sub-diffusive	Rad6, Pep5, and Ufd4 Ub ligases	Homology-driven repair	FROS, time-lapse fluorescence, mass spectrometry analysis, and BIR-PCR
[6]	Yeast (Hap.)	Zeocin (250–300 µg/mL); HO @ MAT locus	increased	increased	Sub-diffusive	Uls1	Homology-driven repair	FROS, time-lapse fluorescence, and BIR-PCR
[90]	Yeast (Hap.)	Endogenous damage; Zeocin (50 µg/mL); I-sceI (2.7 kb from ZWF1 locus)	increased	-	Sub-diffusive	Rad51, Rad54, Mec1, Rad9, Sml1	Homology-driven repair	FROS, time-lapse fluorescence, and recombination and primer-extension assay
[7]	Yeast (Hap.)	Zeocin (250 µg/mL)	-	increased	Sub-diffusive	H2AS129	Enhanced NHEJ; decreased translocation rates	FROS, time-lapse fluorescence, and plasmid repair and translocation assay
[12]	Yeast (Hap.)	Zeocin (100–500 µg/mL)	-	increased	Sub-diffusive	Mec1, Nhp6, Ino80, Rad53	Homology driven repair	FROS, time-lapse fluorescence, super-resolution imaging, genome-wide nucleosome mapping, and recombination assay
[10]	Yeast (Hap.)	Zeocin (250 µg/mL)	-	increased	Sub-diffusive	H2AS129	-	FROS, time-lapse fluorescence, super-resolution imaging, and Langevin dynamics simulations
[89]	Yeast (Hap.)	HO @ MAT loci	increased	-	Sub-diffusive	SWR1, HTZ1	DSB relocalization	FROS and time-lapse fluorescence.
[76]	Yeast (Hap.)	I-sceI (240 kb far from CEN II), Zeocin (250 µg/mL), Phleomycin (3 µg/mL)	increased	increased	Sub-diffusive	Nuclear actin and microtubules	Telomere distribution	FROS, time-lapse fluorescence, and polymer chain simulations
[8]	Yeast (Hap.)	HO @ MAT locus (+TG rich domains)	increased	-	sub-diffusive	Uls1	TG-free mobility: DSB relocalization (NHEJ) TG-rich mobility: translocation	FROS, time-lapse fluorescence, and zoning/translocation assay
[4]	Yeast (Dip.)	I-sceI @ 4 kb from URA3 locus (30 kb from CENV), IR (40 Gy, 200 Gy)	increased	-	Sub-diffusive	Rad51, Sae2	Enhanced homology search.	FROS, time-lapse fluorescence, and genomic blot
[13]	Yeast (Dip., Hap.)	I-sceI @ 4 kb from URA3 locus (30 kb from CENV), IR (40 Gy)	increased	increased	Sub-diffusive	Rad51	Enhanced homology search	FROS, time-lapse fluorescence, and reotation regime model
[63]	Yeast (Hap.)	Zeocin ((50 µg/mL), MMS (0.03%)	-	increased	Super-diffusive	Kir, Tub3, Rad9, Rad52, Rad51, Rad53	DSB relocalization	FROS, time-lapse fluorescence, and BIR-DSB repair efficiency
[91]	Yeast (Hap.)	HO @ MAT locus	decreased	-	Sub-diffusive	Sae2, Ku70	DSB ends tethering	ANCHOR, time-lapse fluorescence, and time-course resection assay
[9]	Yeast (Hap.)	Zeocin (50 µg/mL), MMS (0.03%)	-	increased	Sub-diffusive	INO80, Rad53, Rad9	-	FROS, time-lapse fluorescence
[14]	Yeast (Dip.)	IR (40 Gy)	-	increased	Sub-diffusive	Rad51, Rad52, Mec1/Tel1	Enhanced homology search	FROS, time-lapse fluorescence
[11]	Yeast (Hap.)	HO @ MAT locus	increased	increased	Sub-diffusive	Cep3, Rad53	Dispensable for repair	FROS, time-lapse fluorescence, and HR repair analysis

(Hap.) Haploid; (Dip.) Diploid.

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
