# Peer review of "The Dynamic Behavior of Chromatin in Response to DNA Double-Strand Breaks"

_genes, 2022, doi:10.3390/genes13020215_

Round 1
Reviewer 1 Report
In their manuscript entitled "The dynamic behavior of chromatin in response to DNA double-strand breaks", Garcia Fernandez and Fabre provide an extensive review about chromatin mobility in response to DNA damage and particularly DNA double-strand breaks (DSB). In particular, they provide extensive discussion regarding the mobility of damaged chromatin itself (local mobility) but also the impact of DSB on chromatin which is not directly damaged (global mobility).
The topic is of direct interest for the DSB repair field but this review will also be a good read for anyone interested in chromatin dynamics and nuclear organization. The manuscript is well written and covers most of the interesting recent data in the field.
I have several suggestions which I think can improve the current manuscript.
I also listed the typos and/or formatting issues that I spotted on the submitted manuscript.
Main point:
The review discusses extensively the results obtained in yeast, which is completely justified since a large fraction of the significant papers in the discussed topic where indeed using yeast as a model system. However, I feel that some studies performed in mammals could find their place in the current version of the review without the need for extensive rewriting.
First, I find a bit surprising that recent data regarding the role of cohesin, loop extrusion and Topologically Associating Domains are not included in the review. While it is not clear yet if this is related to global mobility, sufficient data now suggest that cohesin-mediated loop extrusion plays an important part in chromatin movements around DSBs. I suggest this could be done by a short addition in part 5 (Models explaining DSB-induced chromatin mobility) but this could be done in a different way that the authors may find appropriate. Here is a suggested list of reports which could be cited in that context : Natale, Nat Comm, 2017, PMID: 28604675; Ochs, Nature, 2019, PMID: 31645724; Collins, Nat Comm 2020, PMID: 32572033; Saunders, Nat Comm 2020, PMID: 33268790; together with reference 84 of the submitted manuscript.
Similarly, in part 6 (Potential functions of DSB-induced chromatin mobility, subsection 1/ To relocalize DSB), I think the following sentence "Thus, studies in yeast have shown that enhanced local dynamics of a DSB targeted in rDNA enabled its exclusion from the nucleolus [59], [116]" should also include the following work in mammalian systems : van Sluis, Genes Dev, 2015, PMID: 26019174; Harding, Cell Report, 2015, PMID: 26440899, Marnef, Genes Dev 2019, PMID: 31395742.
Finally, in part 2 (The 3Rs of Double Strand Breaks, response, repair, and restoration : an overview), I think that the following sentence "Phosphorylated H2AX (known as γH2AX) serves as a docking site for checkpoint proteins such as 53BP1 that mirrors the phosphorylation mark" is a bit simplistic since 53BP1 recruitment involves many factors downstream of γH2AX. This could be corrected by mentioning reviews on the matter such as Mirma and de Lange, Genes Dev 2020 PMID: 31896689 or Panier and Boulton, Nat Rev Mol Cell Biol, 2014 PMID: 24326623.
list of Typos/formatting issues :
. Abstract :
"yielding/leading". I don't think it is justified to keep both words.
. Part 3. How, where and when to measure undamaged chromatin dynamics?, sub part
. the How :
"modeling yeast chromosomes as generic, semi-flexible polymers with few sequence-specific constraints rigidity". I think it should be replaced by "with a few sequence-specific rigidity constraints"
"Imaging techniques and microscopic tools at at single-molecule resolution have also been emergingsuch as single-particle tracking (SPT)". Repetition of at and emergingsuch not separated.
. The Where :
"for internal chromosomein regions in yeast". chromosomein not separated.
. he When :
"to the involvement of the cohesin complex, which mediates the connection of sister chromatids after replication (Heun et al. 2001;Dion et al. 2013; Cheblal et al. 2020)". Reference formatting issue
. Part 4. Chromatin dynamics as a part of the DNA Damage Response
"Induction of both single and random DSBs was associated with enhanced local mobilit". should be mobility
"than the local mmobility [11]". double m in mobility.
Part 5. Models explaining DSB-induced chromatin mobility
c) Extrinsic forces mediated by the nuclear architecture
"Thus, although changes in the peri-cetromeric region" should be peri-centromeric
Author Response
In their manuscript entitled "The dynamic behavior of chromatin in response to DNA double-strand breaks", Garcia Fernandez and Fabre provide an extensive review about chromatin mobility in response to DNA damage and particularly DNA double-strand breaks (DSB). In particular, they provide extensive discussion regarding the mobility of damaged chromatin itself (local mobility) but also the impact of DSB on chromatin which is not directly damaged (global mobility).
The topic is of direct interest for the DSB repair field but this review will also be a good read for anyone interested in chromatin dynamics and nuclear organization. The manuscript is well written and covers most of the interesting recent data in the field.
I have several suggestions which I think can improve the current manuscript.
I also listed the typos and/or formatting issues that I spotted on the submitted manuscript.
We thank the reviewer for his comments and suggestions that give our review a broader scope. We have taken the comments into account as follows,
Main point:
The review discusses extensively the results obtained in yeast, which is completely justified since a large fraction of the significant papers in the discussed topic where indeed using yeast as a model system. However, I feel that some studies performed in mammals could find their place in the current version of the review without the need for extensive rewriting.
First, I find a bit surprising that recent data regarding the role of cohesin, loop extrusion and Topologically Associating Domains are not included in the review. While it is not clear yet if this is related to global mobility, sufficient data now suggest that cohesin-mediated loop extrusion plays an important part in chromatin movements around DSBs. I suggest this could be done by a short addition in part 5 (Models explaining DSB-induced chromatin mobility) but this could be done in a different way that the authors may find appropriate. Here is a suggested list of reports which could be cited in that context : Natale, Nat Comm, 2017, PMID: 28604675; Ochs, Nature, 2019, PMID: 31645724; Collins, Nat Comm 2020, PMID: 32572033; Saunders, Nat Comm 2020, PMID: 33268790; together with reference 84 of the submitted manuscript.
We thank the referee for this input. Our passage on cohesins in the previous version was a bit brief. We have added information about the formation of chromatin loops and their impact on the dynamics in part 5, page 13 (paragraph 1) and about their function in part 6, page 15 (paragraph 2).
Paragraph 1:
- d) Cohesin-mediated loop extrusion
While it is not clear yet if chromatin loops are related to global mobility, sufficient data now suggest that cohesin-mediated loop extrusion plays an important part in chromatin movements around DSBs. Recent evidences in yeast and mammals have shown that, following damage, chromatin folds into cohesin-dependent loops that influence dynamic inter- and intra-chromosomal contacts [115]–[119]. For example, in yeast, following an HO induced DSB, chromatin flanking the DSB is proposed to prevent cohesion progression along the chromosome. Subsequent folding of chromatin into loops ~20 kb long promotes increased cis contacts while inhibiting trans contacts [118].
Similarly, mammalian TADs are formed by chromatin loops cohesin-mediated. Cohesins are found enriched at TAD boundaries due to their interaction with the border protein CTCF [119]–[122]. CTCF and cohesins form a dynamic fast-exchange complex that makes TADs highly dynamic [121]. Induction of a DSB within a TAD results in loop extrusion triggered by ATM-dependent cohesin accumulation and promotes H2AX phosphorylation during loop extrusion in a directional and progressive manner [119].
Paragraph 2:
2/ To amplify the DNA Damage Response
Increased chromatin dynamics after DSB induction has been shown to increase the local processivity of certain DDR factors, including ATM, 53BP1, RIF1, thereby ensuring efficient DDR where chromatin loop formation may have a role to play [115]–[117], [119], [122]. For example, several groups have observed that local movements within a TAD through chromatin loops bring distant nucleosomes into spatial proximity to ATM kinase, ensuring phosphorylation of γH2AX across the TAD [115], [117], [119], [122]. Therefore, deletion of the CTCF chromatin binding protein reduces both, chromatin movement and γH2AX spreading in a TAD [121], [122]. Furthermore, loop extrusion is not observed in ATM-deficient cells, indicating the presence of a mechanism to protect the integrity of genome structure during DNA damage repair [117] .
Similarly, the ability of yeast Mec1/Tel1 kinases to phosphorylate not only at the DSB, but also in undamaged regions in trans, further supports the importance of chromatin mobility in generating DDR platforms, perhaps through chromatin loops [137], [138]. Accordingly, in trans propagation of γH2A in peri-centromeric regions is consistent with the presence of chromatin loops in centromeric regions, reviewed in [139], although Mec1 propagation in 3D, as reported through mathematical modeling does not necessarily requires loop formation [140].
Similarly, in part 6 (Potential functions of DSB-induced chromatin mobility, subsection 1/ To relocalize DSB), I think the following sentence "Thus, studies in yeast have shown that enhanced local dynamics of a DSB targeted in rDNA enabled its exclusion from the nucleolus [59], [116]" should also include the following work in mammalian systems : van Sluis, Genes Dev, 2015, PMID: 26019174; Harding, Cell Report, 2015, PMID: 26440899, Marnef, Genes Dev 2019, PMID: 31395742.
We have accordingly added the mammals references.
Finally, in part 2 (The 3Rs of Double Strand Breaks, response, repair, and restoration : an overview), I think that the following sentence "Phosphorylated H2AX (known as γH2AX) serves as a docking site for checkpoint proteins such as 53BP1 that mirrors the phosphorylation mark" is a bit simplistic since 53BP1 recruitment involves many factors downstream of γH2AX. This could be corrected by mentioning reviews on the matter such as Mirma and de Lange, Genes Dev 2020 PMID: 31896689 or Panier and Boulton, Nat Rev Mol Cell Biol, 2014 PMID: 24326623.
We have corrected as follows (page 2). Phosphorylated H2AX (known as γH2AX) serves as a docking site for certain remodeling complexes, such as INO80 or SWR1, and checkpoint proteins such as 53BP1. These proteins mirror the phosphorylation mark and in turn can trigger downstream DDR factors [30], [33]–[35], reviewed in [36], [37].
list of Typos/formatting issues :
. Abstract :
"yielding/leading". I don't think it is justified to keep both words.
We have kept the term "leading" as the most appropriate.
. Part 3. How, where and when to measure undamaged chromatin dynamics?, sub part
. the How :
"modeling yeast chromosomes as generic, semi-flexible polymers with few sequence-specific constraints rigidity". I think it should be replaced by "with a few sequence-specific rigidity constraints"
We have corrected accordingly.
"Imaging techniques and microscopic tools at at single-molecule resolution have also been emergingsuch as single-particle tracking (SPT)". Repetition of at and emergingsuch not separated.
We have corrected accordingly.
. The Where :
"for internal chromosomein regions in yeast". chromosomein not separated.
We have corrected accordingly.
. the When :
"to the involvement of the cohesin complex, which mediates the connection of sister chromatids after replication (Heun et al. 2001;Dion et al. 2013; Cheblal et al. 2020)". Reference formatting issue
We have corrected accordingly.
. Part 4. Chromatin dynamics as a part of the DNA Damage Response
"Induction of both single and random DSBs was associated with enhanced local mobilit". should be mobility
"than the local mmobility [11]". double m in mobility.
We have corrected accordingly.
Part 5. Models explaining DSB-induced chromatin mobility
c) Extrinsic forces mediated by the nuclear architecture
"Thus, although changes in the peri-cetromeric region" should be peri-centromeric
We have corrected accordingly.

Reviewer 2 Report
The manuscript from Garcia Fernandez and Fabre reviews chromatin dynamics following induction of double strand breaks and its role in the DNA damage response. The authors present the different types of chromatin mobility (local and global) and discuss the models in the literature to explain the underlying mechanisms.
The manuscript is exhaustive and well structured. I just have few suggestions for its improvement.
Table 1. This table is very informative. Please, modulate column width and font size to avoid inappropriate word breaks. Please, explain the meaning of (H) and (D) in a footnote.
Figure 1. Since you have enough space, you can use full words instead of abbreviations (NEV, MT etc.). Figure legend: I would say “Local and global mobility are symbolised by solid and dashed red arrows (not curves), respectively.
Figures 2 and 3. Both figures are out of focus and the names of the involved factors cannot be read. In Figure 3, the four possible scenarios may be indicated by numbers.
Typing
Page 2. Please, explain the meaning of PTM and TADs
Page 4. Directional change distribution is DCD not CDC, check also in page 7.
Author Response
The manuscript from Garcia Fernandez and Fabre reviews chromatin dynamics following induction of double strand breaks and its role in the DNA damage response. The authors present the different types of chromatin mobility (local and global) and discuss the models in the literature to explain the underlying mechanisms.
The manuscript is exhaustive and well structured. I just have few suggestions for its improvement.
We thank the reviewer for his feedback and his comments, which we have taken into account as follows,
Table 1. This table is very informative. Please, modulate column width and font size to avoid inappropriate word breaks. Please, explain the meaning of (H) and (D) in a footnote.
We have added a foot note, (H) Haploid; (D) Diploid
Figure 1. Since you have enough space, you can use full words instead of abbreviations (NEV, MT etc.).
We have replaced the abbreviations in figure 1 NEV, nuclear envelope; MT, microtubules; CEN, centromere,; TEL, telomere; SPB, spindle pole body.
Figure legend: I would say “Local and global mobility are symbolised by solid and dashed red arrows (not curves), respectively.
We have corrected accordingly.
Figures 2 and 3. Both figures are out of focus and the names of the involved factors cannot be read.
We have enlarged the characters and inserted net figures.
In Figure 3, the four possible scenarios may be indicated by numbers.
We have corrected accordingly.
Page 2. Please, explain the meaning of PTM and TADs
We have explained the abbreviations PTM, Post Translational Modifications; TADs, Topologically Associated Domains
Page 4. Directional change distribution is DCD not CDC, check also in page 7.
We have corrected accordingly.
